# Spatio–Temporal Image Representation of 3D Skeletal Movements for View-Invariant Action Recognition with Deep Convolutional Neural Networks [note 1]

**DOI:** 10.3390/s19081932

**Published:** 2019-04-24

**Authors:** Huy Hieu Pham, Houssam Salmane, Louahdi Khoudour, Alain Crouzil, Pablo Zegers, Sergio A. Velastin

**Affiliations:** 1Cerema, Project team STI, 1 avenue du Colonel Roche, F-31400 Toulouse, France; houssam.salmane@cerema.fr (H.S.); louahdi.khoudour@cerema.fr (L.K.); 2Informatics Research Institute of Toulouse (IRIT), Paul Sabatier University, Toulouse 31062, France; alain.crouzil@irit.fr; 3Aparnix, La Gioconda 4355, 10B, Las Condes, Santiago 7550076, Chile; pablozegers@gmail.com; 4Cortexica Vision Systems Ltd., London SE1 9LQ, UK; sergio.velastin@ieee.org; 5School of Electronic Engineering and Computer Science, Queen Mary University of London, London E1 4NS, UK; 6Department of Computer Science, University Carlos III of Madrid, 28903 Leganés, Spain

**Keywords:** 3D human action recognition, skeleton-based representation, SPMF, Enhanced-SPMF, AHE, D-CNNs, DenseNet

## Abstract

Designing motion representations for 3D human action recognition from skeleton sequences is an important yet challenging task. An effective representation should be robust to noise, invariant to viewpoint changes and result in a good performance with low-computational demand. Two main challenges in this task include how to efficiently represent spatio–temporal patterns of skeletal movements and how to learn their discriminative features for classification tasks. This paper presents a novel skeleton-based representation and a deep learning framework for 3D action recognition using RGB-D sensors. We propose to build an action map called SPMF (*Skeleton Posture-Motion Feature*), which is a compact image representation built from skeleton poses and their motions. An Adaptive Histogram Equalization (AHE) algorithm is then applied on the SPMF to enhance their local patterns and form an enhanced action map, namely Enhanced-SPMF. For learning and classification tasks, we exploit Deep Convolutional Neural Networks based on the DenseNet architecture to learn directly an end-to-end mapping between input skeleton sequences and their action labels via the Enhanced-SPMFs. The proposed method is evaluated on four challenging benchmark datasets, including both individual actions, interactions, multiview and large-scale datasets. The experimental results demonstrate that the proposed method outperforms previous state-of-the-art approaches on all benchmark tasks, whilst requiring low computational time for training and inference.

## 1. Introduction

Human action recognition [1] is one of the most important and challenging tasks in computer vision. Detecting and recognizing correctly what humans do in unknown videos serve as a key component of many real-world applications such as smart surveillance [2,3], human–object interaction [4,5], autonomous vehicle technology [6,7], etc. Although significant progress has been achieved over two decades of research, video-based human action recognition is still a challenging issue due to a number of obstacles, e.g., changes in camera viewpoint, occlusions, background, surrounding distractions, diversity in length and speed of actions [8].

As many other visual recognition tasks, traditional approaches on human action recognition [9] have focused on extracting hand-crafted local features and building local descriptors from RGB sequences provided by 2D cameras. Some typical examples that have been widely exploited with success are SIFT [10,11], HOG/HOF [12,13], HOG-3D [14], Cuboids [15], SURF [16] and Extended SURF [17]. Since these approaches typically recognize actions based on the appearance and movement of the human body parts from a monocular RGB video sequence, they tend to lack 3D structure from the scene. Therefore, single modality human action recognition based only on RGB videos is not enough to overcome the current challenges.

The availability of low-cost and easy-to-use depth sensors such as the Microsoft Kinect™ sensor [18] has helped the computer vision community improve action recognition. These sensors are able to provide detailed 3D structural information of human motion, which is considered complex for traditional 2D cameras. Many action recognition approaches using RGB-D cameras have been proposed and advanced the state-of-the-art [19,20,21,22,23,24,25]. In particular, most of currently depth-sensing cameras have integrated real-time skeleton estimation and tracking frameworks [26,27], helping to facilitate the collection of skeleton sequences. This data source is a high-level representation allowing to describe human action in a more precise and effective way, which is suitable for the problem of action analysis and recognition. Skeleton-based human action recognition is a time-series problem. The skeletal data comprises 3D coordinates of the key joints in the human body over time. This is an effective representation for structured motion [28] because each human action can be represented through the movement of skeleton sequences. Moreover, a large set of actions can be distinguished from these movements [29]. 3D skeletal data is not only invariant to camera-viewpoint but also can be estimated in real-time. Moreover, it is available for most of depth based action datasets [30]. Hence, exploiting this data source for 3D human action recognition opens up opportunities for addressing the limitations of RGB-depth modalities-based solutions and so many skeleton-based action recognition approaches have been proposed [19,23,31,32,33]. Our goal is to exploit the potential of low-cost consumer depth cameras for identifying salient spatio–temporal patterns in skeleton sequences and then explore them for improving the recognition of human actions using deep learning models.

In the literature of skeleton-based action recognition, there are two main issues that need to be solved. The first challenge is to find a skeleton-based representation that transforms the raw skeletal data into a representation that effectively captures the spatio–temporal dynamics of human skeleton joints. The second challenge is to model and recognize actions that are complex, variable and have large intra-class correlation, from the skeleton-based representation. Previous studies [19,23,31,34,35,36,37,38,39,40,41,42] on this topic can be divided into two main categories: skeleton-based action recognition based on hand-crafted features and skeleton-based action recognition using deep neural networks. The first group of methods uses hand-crafted local features and probabilistic graphical models such as Hidden Markov Model (HMM) [43], Conditional Random Field (CRF) [34], or Fourier Temporal Pyramid (FTP) [23] to model and classify actions. However, almost all of these approaches are shallow, data-dependent and require a lot of feature engineering. The second group of methods considers skeletal data as a time-series patterns and proposes the use of Recurrent Neural Networks (RNNs) [44], especially Recurrent Neural Networks with Long Short-Term Memory units (RNN-LSTMs) [45,46] to analyze and model the contextual information contained in the skeleton sequences. They are considered as the most popular deep learning based approach for skeleton-based action recognition and have achieved high-level performance. Although being able to model the long-term temporal of human motion, RNN-LSTMs [45,46] just consider skeleton sequences as a kind of low-level features by feeding raw skeletal data directly into the network input. The huge number of input features makes them complex, time-consuming and may easily lead to overfitting. Nevertheless, almost all of these networks act just as classifiers and do not extract high-level features for recognition tasks [47].

A practical human action recognition system should be able to detect and recognize actions from different viewpoints, robust to noise and operate in real-time. We believe that an efficient and effective representation for 3D human motion plays a decisive role in improving recognition performance. Motivated by the success of our previous work on the SPMF (*Skeleton Posture-Motion Feature*) representation [48] for video-based human action recognition, in this paper we aim to find a new skeleton-based representation and take full advantages in learning highly hierarchical image features of Deep Convolutional Neural Networks (D-CNNs) to build an end-to-end learning framework for 3D human action recognition from skeletal data. Specifically, we propose a new 3D motion representation, termed as Enhanced-SPMF (*Enhanced Skeleton Posture-Motion Feature*). Similar to the SPMF [48], the proposed Enhanced-SPMF has a 2D image structure with three color channels, which is built from a set of spatio–temporal stages, combining 3D skeleton poses and their motions. Moreover, an Adaptive Histogram Equalization (AHE) algorithm [49] is then applied to the color images to enhance their local patterns and generate more discriminative features for classification task. Figure 1 illustrates an overview of the proposed Enhanced-SPMF. To learn image features and recognize action labels from the proposed representation, different D-CNN models based on the DenseNet architecture [50] have been designed and evaluated.

There are five important hypotheses that motivate us to propose a new skeleton-based representation and design DenseNets [50] for 3D human action recognition with skeletal data. First, human actions can be correctly represented through the skeleton movements [28,29]. Second, compared to RGB and depth streams that contain thousands of pixels per frame, skeletal data has a high-level abstraction with much less complexity. This makes the training and inference processes much simpler and faster. Third, as shown in our previous works [48,51], the spatio–temporal dynamics of skeleton sequences can be transformed into color images—a kind of 3D tensor-structured representation that can be effectively learned by representation learning models as D-CNNs. Fourth, many different action classes share a great number of similar primitives, which interferes with action classification. Therefore, extracting essential spatio–temporal patterns from skeleton movements plays a key role in this task. Last, recent research results indicate that CNNs have achieved outstanding performances in many image recognition tasks [52,53]. There are a many signs that seem to indicate that the learning performance of CNNs can be significantly improved by increasing the depth of their architectures [54,55,56,57]. In particular, D-CNNs with architectures such as DenseNet [50] can improve accuracy in the image recognition task since this kind of network is able to prevent overfitting and degradation phenomena [58] by maximizing information flow and facilitating features reuse as each layer in its architecture has direct access to the features from previous layers. Therefore, we explore the use of DenseNet in this work and optimize this architecture for learning and recognizing human actions on the proposed image-based representation.

The effectiveness of the proposed method is evaluated on four public benchmark RGB-D datasets, including MSR Action3D [59], KARD [60], SBU Kinect Interaction [61] and NTU-RGB+D datasets [39]. The hypotheses above were reinforced since the experimental results show that we achieve state-of-the-art performance on all the reported benchmarks. Furthermore, we also report the effectiveness of this approach in terms of computational cost, for both training time and inference latency. Overall, the main contributions of our study include two aspects:Firstly, we present Enhanced-SPMF, a new skeleton-based representation for 3D human action recognition from skeletal data. This work is an extended version of our paper published in the 25th IEEE International Conference on Image Processing (ICIP) [48] in which the Enhanced-SPMF is an extension of SPMF (*Skeleton Pose-Motion Feature*). Compared to our previous work, the current work aims to improve the efficiency of the 3D motion representation via a smoothing filter and a color enhancement technique. The smoothing filter helps us to reduce the effect of noise on skeletal data, meanwhile the color enhancement technique could make the proposed Enhanced-SPMF more robust and discriminative for recognition task. An ablation study on the Enhanced-SPMF demonstrated that the new representation leads to better overall action recognition performance than the SPMF [48].Secondly, we present a deep learning framework (The implementation and models will be made publicly available at https://github.com/cerema-lab/Sensors-2018-HAR-SPMF). based on the DenseNet architecture [50] for learning discriminative features from the proposed Enhanced-SPMF and performing action classification. The framework directly learns an end-to-end mapping between skeleton sequences and their action labels with little pre-processing. We evaluate the proposed method on four highly competitive benchmark datasets and demonstrate significantly improvement over existing state-of-the-art approaches. Our computational efficiency evaluations show that the proposed method is able to achieve high-level of performance whilst requiring low computational time for both the training and inference stages. Compared to our previous work that exploited the Residual Inception v2 network [48], the current work uses a more powerful deep learning model for action recognition task

The rest of this paper is organized as follows: Section 2 discusses related works. Section 3 presents the details of the proposed approach. Datasets and experiments are described in Section 4. The experimental results and analyses are provided in Section 5. Section 6 concludes the paper.

## 2. Related Work

In this section, we briefly review the exiting literature closely related to the topic of deep learning based approaches for 3D human action recognition from skeleton sequences, including skeleton-based action recognition using hand-crafted features and deep learning-based action recognition. We encourage the readers to refer to an extensive review by Han et al. [62] for getting a more comprehensive picture on this topic.

### 2.1. Hand-Crafted Approaches for Skeleton-Based Human Action Recognition

Earlier studies on skeleton-based human action recognition focus on finding well-designed hand-crafted features and using temporal graphical models to analyze the global temporal evolution of skeleton joints. Since when the first work on 3D human action recognition from depth data was introduced [59], many approaches for skeleton-based action recognition have been proposed [19,23,31,34,35,36]. The common characteristic of these approaches is that, they extract geometric features of 3D joint movements and model their temporal information by a generative model. For instance, Wang et al. [19] represented the human motion by means of the pairwise relative positions of the skeleton joints for generating more discriminative features. Fourier Temporal Pyramid (FTP) [19] was then proposed to model the temporal dynamics of the actions from LOPs. Vemulapalli et al. [23] represented the 3D geometric relationships of body parts as points in a Lie Group and then exploited Dynamic Time Warping (DTW) [63] and Fourier Temporal Pyramid (FTP) [19] to model their temporal dynamics. Xia et al. [31] extracted and computed histograms of 3D joint locations (HOJ-3D) to represent actions via posture visual words. The temporal evolutions of those words are modeled by a discrete Hidden Markov Models (HMM) [64]. Instead of modeling temporal evolution of skeletons, Luo et al. [35] proposed a discriminative dictionary learning algorithm (called DL-GSGC) that incorporated both group sparsity and geometry constraints to learn motion features from the 3D joint positions. An encoding technique called Temporal Pyramid Matching (TPM) [35] was then used for keeping the temporal information and performing action classification.

Although promising results have been achieved, the above approaches have some limitations that are difficult to overcome. For instance in many cases, they require pre-processing input data in which the skeleton sequences need to be segmented or aligned. Unlike these approaches, we propose a skeleton-based representation and a deep learning framework for 3D human action recognition that learns to recognize actions directly from the original skeletons in an end-to-end manner, without dependence on the length of actions. Moreover, the proposed solution is general and can be applied with some other data modalities such as motion capture data [65] and the output of pose estimation algorithms [66,67].

### 2.2. Deep Learning Approaches for Skeleton-Based Human Action Recognition

Approaches based on Recurrent Neural Network with Long Short-Term Memory units (RNN-LSTM) [45,68] are the most popular deep learning approach for skeleton-based action recognition and have achieved high-level performance for video-based action recognition tasks [37,38,39,40,41,42]. The temporal evolutions of skeletons are spatio–temporal patterns. Thus, they can be modeled by memory cells in the structure of RNN-LSTMs [45,68]. For instance, Du et al. [37] proposed to use a hierarchical RNN to model the long-term contextual information of skeletal data, in which the human skeleton was divided into five parts according to its physical structure. Each low-level part was modeled by an RNN and then combined into the final representation of high-level parts for action classification. Shahroudy et al. [39] introduced a part-aware LSTM human action learning model by splitting a long-term memory of the entire motion to part-based cells. The long-term context of each body part was learned independently. The output of the network was then formed as a combination of independent body part context information. Liu et al. [40] presented a spatio–temporal LSTM network, called ST-LSTM, for 3D action recognition from skeletal data. They proposed a skeleton-based tree traversal technique to feed the structure of the skeletal data into a sequential LSTM network and improved the performance of the ST-LSTM by adding more trust gates. Recently, Liu et al. [42] focused on selecting the most informative skeleton joints by using a new class of LSTM network, namely Global Context-Aware Attention LSTM (GCA-LSTM), for 3D skeleton-based action recognition. Two LSTM layers were used. The first layer encodes the input sequences and generates an initial global context memory for these sequences. Meanwhile, the second layer performs attention over the input sequences with the assistance of obtained global context memory. The attention representation was then used back to refine the global context. Multiple attention iterations are executed and the final global contextual information is used for action classification task.

Compared to the approaches based on hand-crafted local features, the RNN-LSTM based approaches and their variants have been showing superior action recognition performance. However, they tend to overemphasize the temporal information and lose the spatial information of skeletons [37,38,39,40,41,42]. RNN-LSTM based approaches still struggle to cope to scope with the complex spatio–temporal variations of skeletal movements due to a number of issues such as jitters and movement speed variability. Another drawback of the RNN-LSTM networks [45,68] is that they just model the overall temporal dynamics of actions without considering the detailed temporal dynamics of them. To overcome these limitations, we propose in this study a CNN-based approach that is able to extract discriminative features of actions and model various temporal dynamics of skeleton sequences via the proposed Enhanced-SPMF representation, including both short-term, medium-term, and long-term actions. We summarize the advantages and disadvantages of our proposed method in comparison with some previous approaches in Table 1.

## 3. Method

The details of the proposed approach are presented in this section. Figure 2 illustrates the key components of the proposed learning framework for recognizing actions from skeleton sequences. We first show how skeleton pose and motion features can be combined to build an action map in the form of an image-based representation (Section 3.1), and how to use a color enhancement technique for improving the discriminative ability of the proposed representation (Section 3.2). We then introduce an end-to-end deep leaning framework based on DenseNets to learn and classify actions from the enhanced representations (Section 3.3). Before that, in order to put the proposed approach into context, it is useful to review the central ideas behind the original DenseNet architecture (Section 3.3.1).

### 3.1. SPMF: Building Action Map from Skeletal Data

One of the major challenges in exploiting D-CNNs for skeleton-based action recognition is how the spatio–temporal patterns of skeleton movements could be effectively represented and fed to D-CNNs for representation learning. As D-CNNs work well on image representations [73], our idea therefore is to encode the whole skeleton sequence into a single 2D image as a global representation for the action sequence. In general, two essential elements that determine a human action are poses and their motions. Hence, we decide to transform these two important elements into the static spatial structure of a color image with three *R*, *G*, *B* channels. Specifically, we propose a new representation, namely Enhanced-SPMF (*Enhanced Skeleton Pose-Motion Feature*), which is built from pose and motion vectors extracted from the skeleton joints. Note that, combining multiple kinds of geometric features such as joint coordinates, lines and planes determined by the joints will lead to lower performance than using only a single type of feature or several main type of features [74]. Moreover, it has been reported [61] that joint features such as joint-joint distance and joint-joint motion are the strongest features among many others.

#### 3.1.1. Pose Features (PFs) Computation

Given a skeleton sequence S with *N* frames, denoted by $S=\{F^{t}\}$, where t=1,2,3,…,N. Let $p_{j}^{t}$ and $p_{k}^{t}$ be the 3D coordinates of the *j*-th and *k*-th joints in $F^{t}$. The Joint-Joint Distance $JJD_{jk}^{t}$ between $p_{j}^{t}$ and $p_{k}^{t}$ at timestamp *t* is computed as
(1)$$JJD_{jk}^{t}=\left|\right|p_{j}^{t}-p_{k}^{t}{\left|\right|}_{2},\;\left(t=1,2,3,\ldots ,N\right),$$
where $\left|\right|\cdot {\left|\right|}_{2}$ denotes the Euclidean distance between two joints. The joint distances obtained by Equation (Equation 1) for all types of actions of a specific dataset range from $D_{\min}=0$ to $D_{\max}=\max\left\{JJD_{jk}^{t}\right\}$. We note this distance space as $D_{original}$. In fact, $D_{original}$ can be transformed into a tensor-structure and fed directly to D-CNNs for learning action features. However, since $D_{\mathrm{original}}$ is a high-dimensional space, it could lead D-CNNs to overfit as well as being time-consuming. Thus, we need to describe the input skeleton sequences as low-dimensional signals such that they are easy to parameterize by learning models and discriminative enough for a classification task. To do that, we normalize all elements of $D_{original}$ to the range [0,1], denoted as $D_{\left[0,1\right]}$. To reflect the change in joint distances, we encode $D_{\left[0,1\right]}$ into a color space using a sequential discrete color palette called JET color map (A JET color map is based on the order of colors in the spectrum of visible light, ranging from blue to red, and passing through the cyan, yellow, and orange.). The encoding process converts the joint distances $JJD_{jk}^{t}\in D_{\left[0,1\right]}$ for all possible combinations *j* and *k* into color points ${\mathrm{JJD}}_{RGB}^{t}\in N_{\left[0,255\right]}^{3}$ performed by 256-color JET scale. To this end, we first normalize the distance values with respect to the maximum and minimum values of a grayscale image ranging from 0 to 1. As illustrated in Figure 3, the scalar distances are converted to a three channel map via a JET mapping. This technique is similar to depth encoding method presented in [75]. The use of a discrete color palette allows us to reduce complexity of input features. This helps accelerate the convergence rate of deep learning networks during the training stage. Moreover, it should be noted that point-point distances are invariant when they are moved into a new coordinates system in the 3D Euclidean space. Therefore, the use of the Joint-Joint Distance $JJD_{jk}^{t}$ can help our final representation be more independent to the camera viewpoint.

Apart from the distance information, the orientation between joints is also important for describing human motions. The Joint-Joint Orientation ${\mathrm{JJO}}_{jk}^{t}$ from joint $p_{j}^{t}$ to $p_{k}^{t}$ at time-stamp *t* is computed as
(2)$${\mathrm{JJO}}_{jk}^{t}=p_{j}^{t}-p_{k}^{t},\;\left(t=1,2,3,\ldots ,N\right).$$

The ${\mathrm{JJO}}_{jk}^{t}$ is a vector where all of its components *p* can be normalized to the range [0,255]. This can be done via the following transformation
(3)$$p_{\mathrm{norm}}=\mathrm{floor}\left(255\times \frac{p-c_{\min}}{c_{\max}-c_{\min}}\right),$$
where $p_{\mathrm{norm}}$ indicates the normalized value, $c_{\max}$ and $c_{\min}$ are the maximum and minimum values of all coordinates over the training set, respectively. The function floor(·) rounds down to the nearest integer. We consider three components (x,y,z) of ${\mathrm{JJO}}_{jk}^{t}$ after normalization as the corresponding three components (R,G,B) of a color pixel and build ${\mathrm{JJO}}_{RGB}^{t}$ as a 3D array that is formed by all ${\mathrm{JJO}}_{jk}^{t}$ values. We then define “*a human pose*” at timestamp *t* by vector ${\mathrm{PF}}^{t}$ that describes the distance and orientation relationship between skeleton joints,
(4)$$\;{\mathrm{PF}}^{t}=\left[{\mathrm{JJD}}_{RGB}^{t}⧺{\mathrm{JJO}}_{RGB}^{t}\right],\;\left(t=1,2,3,\ldots ,N\right).$$

Here the symbol (⧺) horizontally concatenates vectors ${\mathrm{JJD}}_{RGB}^{t}$ and ${\mathrm{JJO}}_{RGB}^{t}$ together.

#### 3.1.2. Motion Features (MFs) Computation

Let $p_{j}^{t}$ and $p_{k}^{t+1}$ denote the 3D coordinates of the *j*-th and *k*-th joints at two consecutive frames $F^{t}$ and $F^{t+1}$. Similarly to $JJD_{jk}^{t}$ in Equation (Equation 1), the Joint-Joint Distance $JJD_{jk}^{t,t+1}$ between $p_{j}^{t}$ and $p_{k}^{t+1}$ is computed as
(5)$$JJD_{jk}^{t,t+1}=\left|\right|p_{j}^{t}-p_{k}^{t+1}{\left|\right|}_{2},\;\left(t=1,2,3,\ldots ,N-1\right).$$

Also, similarly to Equation (Equation 2), the Joint-Joint Orientation ${\mathrm{JJO}}_{jk}^{t,t+1}$ from joint $p_{j}^{t}$ to $p_{k}^{t+1}$ is computed as
(6)$${\mathrm{JJO}}_{jk}^{t,t+1}=p_{j}^{t}-p_{k}^{t+1},\;\left(t=1,2,\ldots ,N-1\right).$$

We define “*a human motion*” from *t* to t+1 by vector ${\mathrm{MF}}^{t\to t+1}$, in which
(7)$${\mathrm{MF}}^{t\to t+1}=\left[{\mathrm{JJD}}_{RGB}^{t,t+1}⧺{\mathrm{JJO}}_{RGB}^{t,t+1}\right],\;\left(t=1,2,\ldots ,N-1\right),$$
where ${\mathrm{JJD}}_{RGB}^{t,t+1}$ and ${\mathrm{JJO}}_{RGB}^{t,t+1}$ are encoded to qualify the color representation as ${\mathrm{JJD}}_{RGB}^{t}$ and ${\mathrm{JJO}}_{RGB}^{t}$, respectively.

#### 3.1.3. Building Global Action Map from PFs and MFs

Based on the obtained PFs and MFs, we propose a skeleton-based representation called SPMF for 3D human action recognition. To this end, all PFs and MFs computed from the skeleton sequence S are concatenated into a single feature vector in temporal order from the beginning to the end of the action. It is a global representation for the whole skeleton sequence S without dependence on the range of action and can be obtained by
(8)$$\begin{array}{c} \mathrm{SPMF}=[{\mathrm{PF}}^{1}⧺{\mathrm{MF}}^{1\to 2}⧺{\mathrm{PF}}^{2}⧺\ldots ⧺{\mathrm{PF}}^{t}⧺{\mathrm{MF}}^{t\to t+1}⧺{\mathrm{PF}}^{t+1}\ldots ⧺{\mathrm{PF}}^{N-1}⧺{\mathrm{MF}}^{N-1\to N}⧺{\mathrm{PF}}^{N}]. \end{array}$$

Figure 4 (*top row*) shows some SPMFs obtained from the MSR Action3D dataset [59] in which all images are resized to 32 × 32 pixels. Before computing the SPMF, a Savitzky-Golay smoothing filter [37,76] is adopted to reduce the effect of noise on skeletal data. In the experiments, we use the filter
(9)$$\begin{array}{c} f^{t}=\frac{-3c^{t-2}+12c^{t-1}+17c^{t}+12c^{t+1}-3c^{t+2}}{35}, \end{array}$$
where $c^{t}$ denotes the skeleton joint coordinates of frame $F^{t}$ (t=1,2,…,N) and $f^{t}$ denotes the filtering result. This filter design method is described in detailed in Appendix A.

### 3.2. Enhanced-SPMF: Building Enhanced Action Map

The skeleton-based representations obtained by Equation (Equation 8) mainly reflect the spatio–temporal distribution of skeleton joints. We visualize these representations and observe that they tend to be low contrast images, as shown in Figure 4 (*top row*). In this case, a color enhancement method can be useful for increasing contrast and highlighting the texture and edges of the motion maps. Therefore, it is necessary to enhance the local features on the generated color images after encoding. The Adaptive Histogram Equalization (AHE) [49] is a common approach for this task. This technique is capable of enhancing the local features of an image. Mathematically, let I be a given digital image, represented as a *r*-by-*c* matrix of integer pixels with intensity levels in the range [0,L−1]. The histogram of image I will be defined by
(10)$$H_{k}=n_{k},$$
where $n_{k}$ is the number of pixels in I with intensity *k*. The probability of occurrence of intensity level *k* in I can be estimated by
(11)$$p_{k}=\frac{n_{k}}{r\times c},\;\left(k=0,1,2,\ldots ,L-1\right).$$

The histogram equalized image is defined by transforming the pixel intensities, *n*, of I by the function
(12)$$T\left(n\right)=\mathrm{floor}(\left(L-1\right)\sum_{k=0}^{n}p_{k}),\;\left(n=0,1,2,\ldots ,L-1\right),$$

The Histogram Equalization (HE) method is used for increasing the global contrast of the image. However, it cannot solve the problem of increasing local contrast. To overcome this limitation, the image needs to be divided into R regions and the HE is then applied in each and every one of these regions. This technique is called the Adaptive Histogram Equalization algorithm (AHE) [49]. The bottom row of Figure 4 shows samples of the enhanced motion map with R = 8 on 32 × 32 images, which we refer to it as Enhanced-SPMF, for some actions from the MSR Action 3D dataset [59].

### 3.3. Deep Learning Model

#### 3.3.1. Densely Connected Convolutional Networks

DenseNet [50], considered as the current state-of-the-art CNN architecture, has some interesting properties. In this architecture [50], each layer is connected to all the others within a dense block and all layers can access to the feature maps from their preceding layers. Besides, each layer receives direct information flow from the loss function through the shortcut connections. These properties help DenseNet [50] to be less prone to overfitting for supervised learning problems. Mathematically, traditional CNN architectures, e.g., AlexNet [52] or VGGNet [54] connect the output feature maps $x_{l-1}$ of the (l−1)th layer as input to the lth layer and try to learn a mapping function
(13)$$x_{l}=H_{l}(x_{l-1}),$$
where $H_{l}\left(\cdot \right)$ is a non-linear transformation and usually implemented via a series of operations such as Convolution (Conv.), Rectified Linear Unit (ReLU) [77], Pooling [78], and Batch Normalization (BN) [79]. When increasing the depth of the network, the network training process becomes complex due to the vanishing-gradient problem and the degradation phenomenon [58] (please see Appendix B for more details). To solve these problems, He et al. introduced ResNet [56]. The key idea behind the ResNet architecture [56] is the presence of shortcut connections that bypass the non-linear transformations $H_{l}\left(\cdot \right)$ with an identity function id(x)=x. This way, each ResNet building block [56] produces a feature map $x_{l}$ by performing the following computation
(14)$$x_{l}=H_{l}\left(x_{l-1}\right)+x_{l-1}.$$

Inspired by the philosophy of ResNet [56], to maximize information flow through layers, Huang et al. proposed DenseNet [50] with a simple connectivity pattern: the lth layer in a dense block receives the feature maps of all preceding layers as inputs. That means
(15)$$x_{l}=H_{l}\left(\left[x_{0}⧺x_{1}⧺x_{2}⧺\ldots ⧺x_{l-1}\right]\right),$$
where $\left[x_{0}⧺x_{1}⧺x_{2}⧺\ldots ⧺x_{l-1}\right]$ is a single tensor constructed by concatenation of the previous layer’s output feature maps. Additionally, all layers in the architecture receive direct supervision signals from the loss function through the shortcut connections. In this manner, the network is easy to optimize and resistant to overfitting. In DenseNet [50], multiple dense blocks are connected via transition layers. Each transition layer consists of a convolutional layer followed by an average pooling layer that changes the size of feature maps (The concatenation operation used in Equation (Equation 15) is not viable when the size of feature maps changes.). Each block with its transition layer produces *k* feature maps and the parameter *k* is called as the *“growth rate”* of the network. The non-linear function $H_{l}\left(\cdot \right)$ in the original work [50] is a composite function of three consecutive operations: BN-ReLU-Conv.

#### 3.3.2. Network Design

We propose to design and optimize deep DenseNets [50] for learning and classifying human actions on the Enhanced-SPMFs. To study how recognition performance varies with architecture size, we explore different network configurations. The following configurations are used in our experiments: DenseNet (*L* = 100, *k* = 12); DenseNet (*L* = 250, *k* = 24); and DenseNet (*L* = 190, *k* = 40), where *L* is the depth of the network and *k* is the network growth rate. On all datasets, we use three dense blocks on 32×32 input images. In this design, $H_{l}\left(\cdot \right)$ is defined as Batch Normalization (BN) [79], followed by an advanced activation layer called Exponential Linear Unit (ELU) [80] and 3×3 Convolution (Conv.). A Dropout [80] with a rate of 0.2 is used after each Convolution to prevent overfitting. After the feature extraction stage, a Fully Connected (FC) layer is used for classification task in which the number of neurons for this FC layer is equal to the number of action classes in each dataset. The proposed networks can be trained in an end-to-end manner by gradient descent using Adam update rule [81]. During the training stage, we minimize a cross-entropy loss function, which is measured by the difference between the true action label *y* and the predicted action $\hat{y}$ by the networks over the training samples X. In other words, the network will be trained to solve the following optimization problem
(16)$$\mathrm{Arg}\;\min_{W}\;\left(L_{X}\left(y,\hat{y}\right)\right)=\mathrm{Arg}\;\min_{W}\;\left(-\frac{1}{M}\sum_{i=1}^{M}\sum_{j=1}^{C}y_{ij}\log{\hat{y}}_{ij}\right),$$
where W is the set of weights that will be optimized by the model, *M* denotes the number of samples in training set X and *C* is the number of action classes.

## 4. Experiments

We investigate the effectiveness of the proposed approach using four public benchmark action recognition datasets, comparing our method with current state-of-the-art models for each benchmark. We refer the reader to a survey by Zhang et al. [30] for a full description of current RGB-D based action recognition datasets: MSR Action3D [59], KARD [60], SBU Kinect Interaction [61], NTU-RGB+D [39]. The detailed description of each dataset is provided in Section 4.1. The implementation and training methodology are described in Section 4.2.

### 4.1. Datasets and Settings

MSR Action3D dataset [59]: This Kinect 1 captured dataset contains 20 actions performed by 10 subjects. Each skeleton is composed of 20 joints. The MSR Action3D dataset [59] is challenging due to its high inter-action similarities. There are 567 action sequences in total, however, 10 sequences are not valid since the skeletons were missing. Thus, our experiments were conducted on 557 valid sequences. We follow the standard protocol proposed by Li et al. [59]. Specifically, the whole dataset is divided into three subsets: AS1, AS2 and AS3. Table 2 provides a list of actions in each subset, in which all subjects with IDs 1, 3, 5, 7, 9 are selected for training and the remaining subjects with IDs 2, 4, 6, 8, 10 are used for test. Very deep neural networks such as the deep DenseNet architecture require a lot of data to train and optimize. Unfortunately, there are only 557 skeleton sequences on the MSR Action3D dataset [59]. Therefore, some data augmentation techniques, i.e., random cropping, vertical flipping, and rotation with $\alpha =90^{\circ }$ have been applied on this dataset to minimize overfitting. Figure 5 illustrates three data augmentation techniques that were used in our experiments.

Kinect Activity Recognition Dataset (KARD) [60]: The KARD [60] is a Kinect 1 dataset that contains 18 actions and 540 video sequences in total. Each action is performed three times by 10 subjects. It is composed of RGB, depth and skeleton frames in which each skeleton frame contains 15 key joints. The authors of the dataset [60] proposed to divide it into three subsets (i.e., Action Set 1, Action Set 2, and Action Set 3), as listed in Table 3. For each subset, three experiments have been proposed. Specifically, the first experiment (Experiment A) uses one-third of the dataset for training and the rest for test. Meanwhile, the second experiment (Experiment B) uses two-thirds of the dataset for training and the rest for test. The last experiment (Experiment C) uses half of the dataset for training and the other half for testing. As was the case for MSR Action3D dataset [59], data augmentation techniques (i.e., random cropping, vertically flipping, and rotation with $\alpha =90^{\circ }$) were also applied.

SBU Kinect Interaction dataset [61]: This dataset was collected using the Kinect v1 sensor. It contains 282 skeleton sequences and 6822 frames performed by 7 participants. Each frame of the SBU Kinect dataset [61] contains skeleton joints of two subjects corresponding to an interaction, each skeleton has 15 key joints. There are 8 interactions in total, including *approaching, departing, pushing, kicking, punching, exchanging objects, hugging,* and *shaking hands*. This dataset is challenging due to the fact that the joint coordinates exhibit low accuracy. Moreover, they contain non-periodic actions as well as very similar body movements. For instance, there are some pairs of actions that are difficult to distinguish such as *exchanging objects*–*shaking hands* or *pushing*–*punching*. We randomly split the whole dataset into 5 folds, in which 4 folds are used for training and the remaining 1 fold is used for test. It should be noted that each skeleton frame provided by the SBU dataset [61] contains two separate subjects. Therefore, we consider them as two data samples and feature computation is conducted separately for the two skeletons. Additionally, data augmentation (i.e., random cropping, vertically flipping, rotation with $\alpha =90^{\circ }$) has been also applied on the SBU dataset [61].

NTU-RGB+D dataset [39]: This Kinect 2 captured dataset is a very large-scale RGB-D dataset. To the best of our knowledge, the NTU-RGB+D dataset [39] is currently the largest and state-of-the-art benchmark dataset with skeletal data for human action analysis. It provides more than 56 thousand video samples, 4 million frames, collected from 40 distinct subjects for 60 different action classes. The following actions are provided by the NTU-RGB+D dataset [39] (please see Figure 6 for some examples): *drinking, eating, brushing teeth, brushing hair, dropping, picking up, throwing, sitting down, standing up, clapping, reading, writing, tearing up paper, wearing jacket, taking off jacket, wearing a shoe, taking off a shoe, wearing on glasses, taking off glasses, putting on a hat/cap, taking off a hat/cap, cheering up, hand waving, kicking something, reaching into self pocket, hopping, jumping up, making/answering a phone call, playing with phone, typing, pointing to something, taking selfie, checking time, rubbing two hands together, bowing, shaking head, wiping face, saluting, putting palms together, crossing hands in front. sneezing/coughing, staggering, falling down, touching head, touching chest, touching back, touching neck, vomiting, fanning self. punching/slapping other person, kicking other person, pushing other person, patting others back, pointing to the other person, hugging, giving something to other person, touching other persons pocket, handshaking, walking towards each other,* and *walking apart from each other*. In the NTU-RGB+D dataset [39], each skeleton contains the 3D coordinates of 25 body joints. The authors [39] of this dataset suggested two different evaluation criteria, including Cross-Subject evaluation and Cross-View evaluation. For the Cross-Subject setting, the sequences performed by 20 subjects (with IDs 1, 2, 4, 5, 8, 9, 13, 14, 15, 16, 17, 18, 19, 25, 27, 28, 31, 34, 35, and 38) are used for training and the rest sequences are used for test. In Cross-View setting, the sequences provided by cameras 2 and 3 are used for training while sequences from camera 1 are used for test. This setting allows to evaluate the ability to recognize actions under multiple-viewpoints of the proposed skeleton-based representation. We do not apply any data augmentation technique on the NTU-RGB+D [39] due to the very large-scale nature of this dataset [39].

### 4.2. Implementation Details

For all the datasets, the proposed Enhanced-SPMF representations are computed directly from the raw skeleton sequences without using a fixed number of frames. For computational efficiency, all the image representations are resized to 32 × 32 pixels. The three network configurations: DenseNet (*L* = 100, *k* = 12); DenseNet (*L* = 250, *k* = 24); and DenseNet (*L* = 190, *k* = 40) were implemented and evaluated in Python with the support of the Keras framework using TensorFlow as back-end. During the training stage, we use mini-batches of 32 images for all networks. The weights are initialized as per the He initialization technique [82]. Adam optimizer [81] is used with default parameters (i.e., $\beta_{1}=0.9$ and $\beta_{2}=0.999$). Additionally, we use a dynamic learning rate during training. The initial learning rate is set to 0.01 and is decreased by a factor of 0.1 after every 50 epochs. All networks are trained for 300 epochs from scratch.

## 5. Experimental Result and Analysis

### 5.1. Results and Comparisons with the State-of-the-Art

Results on MSR Action3D dataset: Experimental results and comparisons of the proposed method with the current state-of-the-art approaches on the MSR Action3D dataset [59] are summarized in Table 4. We compare the proposed method with Bag of 3D Points [59], Depth Motion Maps [69], Bi-LSTM [72], Lie Group Representation [23], FTP-SVM [72], Hierarchical LSTM [37], ST-LSTM Trust Gates [40], Graph-Based Motion [36], ST-NBNN [70], ST-NBMIM [83], S-T Pyramid [84], Ensemble TS-LSTM v2 [71] and our previous model SPMF Inception-ResNet-222 [48] using the same evaluation protocol. The proposed DenseNets (*L* = 100, *k* = 12) and DenseNet (*L* = 190, *k* = 40) achieve average accuracies of 98.76% and 98.94%, respectively. Meanwhile, the best recognition accuracies are obtained by the proposed DenseNet (*L* = 250, *k* = 24) with a total average accuracy of 99.10%. This result outperforms many previous approaches [23,36,37,40,59,69,70,72,83,84], demonstrating the superiority of the proposed method. Figure 7 (*first row*) shows learning curves of the proposed DenseNets on the AS1 subset/MSR Action3D dataset [59]. The recognition accuracy for each action class in the AS1 subset by the DenseNet (*L* = 250, *k* = 24) is provided in Figure 8 via its confusion matrix.

Results on KARD dataset: We performed a total of 9 experiments over three experiments A, B, and C on the KARD dataset [60]. Table 5 summarizes the obtained results on this dataset. We compute the average recognition accuracy over the three experiments and compare it with existing techniques including Hand-crafted Features [60], Posture Feature+Multi-class SVM [85], and Key Postures+Multi-class SVM [86]. As can be seen in Table 5, the proposed DenseNet (*L* = 250, *k* = 24) is able to improve state-of-the-art accuracy by 9.15% over Hand-crafted Features [60], 2.78% over Posture Feature+Multi-class SVM [85] and 0.68% over Key Postures+Multi-class SVM [86]. This result confirms that the proposed deep learning framework trained on the Enhanced-SPMFs is able to achieve better performance in the recognition of actions compared to hand-crafted based approaches.

Results on SBU Kinect Interaction dataset: As reported in Table 6, the proposed DenseNet (*L* = 250, *k* = 40) achieved an accuracy of 97.86% and outperforms many existing state-of-the-art approaches including Raw Skeleton [61], Joint Features [61], HBRNN [37], CHARM [87], Deep LSTM [41], Joint Features [88], ST-LSTM [40], Co-occurrence+Deep LSTM [41], STA-LSTM [89], ST-LSTM+Trust Gates [40], ST-NBMIM [83], Clips+CNN+MTLN [90], Two-stream RNN [91], and GCA-LSTM network [92]. Using only skeleton modality, the proposed method outperforms hand-crafted feature based approaches such as Raw Skeleton [61], Joint Features [61] and recent state-of-the-art RNN-based approaches [37,40,41,89,91,92]. In particular, the proposed method achieves a significant accuracy gain of 2.96% compared to the nearest competitor GCA-LSTM network [92]. This result demonstrates that the proposed deep learning framework is able to learn discriminative spatio–temporal features of skeleton joints containing in the proposed motion representation for classification task.

Results on NTU-RGB+D dataset: For the NTU-RGB+D dataset [39], the best configuration DenseNet (*L* = 250, *k* = 40) achieves an accuracy of 80.11% on the Cross-Subject evaluation and 86.82% on the Cross-View evaluation, as summarized in Table 7. These results demonstrate the effectiveness of the proposed representation and deep learning framework since they surpass previous state-of-the-art techniques such as Lie Group Representation [23], Hierarchical RNN [37], Dynamic Skeletons [93], Two-Layer P-LSTM [39], ST-LSTM Trust Gates [40], Geometric Features [74], Two-Stream RNN [91], Enhanced Skeleton [94], Lie Group Skeleton+CNN [95], and GCA-LSTM [92]. The experimental results have also shown that the proposed method leads to better overall action recognition performance than our previous models including Skeleton-based ResNet [51] and SPMF Inception-ResNet-222 [48]. With a high recognition rate on the Cross-View evaluation (86.82%) where the sequences provided by cameras 2 and 3 are used for training and sequences from camera 1 are used for test, the proposed method shows its effectiveness for dealing with view-independent action recognition problem. Figure 7 (*last row*) shows the training loss and test accuracy of the DenseNet (*L* = 250, *k* = 24) on this dataset.

### 5.2. An Ablation Study on the Proposed Enhanced-SPMF Representation

We believe that the use of the AHE algorithm [49] and the Savitzky-Golay smoothing filter [37,76] helps the proposed representation to be more discriminative, which improves recognition accuracy. To verify this hypothesis, we carried out an ablation study on the Enhanced-SPMF representation provided by the SBU Kinect Interaction dataset [61]. Specifically, we trained the proposed DenseNet (L=250,k=24) on both the SPMFs and Enhanced-SPMFs. During training, the same hyper-parameters and training methodology were applied. The experimental results indicate that the proposed deep network achieves better recognition accuracy when trained on the Enhanced-SPMFs. As reported in Figure 9, applying the AHE algorithm [49] and and the Savitzky–Golay smoothing filter [37,76] helps improving the accuracy by 4.09%. This result validates our hypothesis above.

### 5.3. Visualization of Deep Feature Maps

Different action classes have different discriminative characteristics. To better understand the internal operation of the proposed deep networks and to study what they learned from the skeleton-based representation, we input different Enhanced-SPMFs corresponding to different action classes of the MSR Action3D dataset [59] to the DenseNet (L=100, k=12) and visualize the individual feature maps learned by the network at the end of a dense block (intermediate layer). We observe that the designed network is able to extract discriminative features from the Enhanced-SPMF representations. This is expressed through the color of each learned feature map, as can be seen in Figure 10. These discriminative features play a key role in classifying actions.

### 5.4. Computational Efficiency Evaluation

In this section, we take the AS1 subset of MSR Action3D dataset [59] and the DenseNet (*L* = 100, *k* = 12) to evaluate the computational efficiency of the proposed method. Figure 11 illustrates three main stages of the deep learning framework for learning and recognizing actions from skeleton sequences, including an encoding process from input skeleton sequences to color images (Stage 1), a supervised training stage (Stage 2), and an inference stage (Stage 3). The implementation is in Python/Keras and when training on a single GeForce GTX 1080 Ti GPU, the proposed deep network only has 6.0M parameters and it takes less than six hours to reach convergence. Latency in predicting an action for a new skeleton sequence (including encoding it to color images, executed on a CPU) is about $74.8\times 10^{-3}$ seconds per sequence. Additionally, it should be noted that the computation of the Enhanced-SPMFs can be implemented and optimized on a GPU for real-time applications. Please see Table 8 for further details. This result verifies the effectiveness of the proposed learning framework in terms of computational cost.

### 5.5. Limitations

The use of the Savitzky-Golay filter [76] helps reduce the effect of noise on the raw skeleton sequences. However, the proposed approach cannot overcome the problem of missing data. In other words, as the Enhanced-SPMF is a global representation for the whole skeleton sequence, data errors of local fragments in the input sequences could reduce the recognition rate. Another open problem of the proposed approach is how to scope with Online Action Recognition (OAR) task. Specifically, how to detect and recognize human actions from unsegmented streams in a continuous manner, where boundaries between different kinds of actions within the stream are unknown. A common solution for OAR is the sliding window based methods [97,98]. These approaches consider the temporal coherence within the window for prediction. We can also apply this idea to solve the current problem. E.g., during the online inference phase, we use a sliding window on the original skeleton sequences or on image-coded representations (i.e., Enhanced-SPMFs) and then predicting action by pretrained deep learning model, as we showed in Figure 11 (Stage 3). However, we understand that the performance of this approach is sensitive to the window size. Either too large or too small window size could lead to a significant drop in recognition performance. Another solution is to use Temporal Attention Networks (TANs) [99,100,101,102] that incorporate temporal attention models for video-based action recognition.

## 6. Conclusions

In this paper, we have presented an efficient and effective deep learning framework for 3D human action recognition from skeleton sequences. A novel motion representation, termed Enhanced-SPMF, which captures the spatio–temporal information of skeleton movements and transforms them into color images has been proposed. We exploited the Adaptive Histogram Equalization (AHE) technique to enhance the local textures of color images and generate more discriminative features for learning and classification tasks. Different Deep Convolutional Neural Networks (D-CNNs) based on the DenseNet architecture have been designed and optimized to learn and recognize actions from the proposed representation, in an end-to-end manner. Extensive empirical evaluations on four challenging public datasets demonstrate the effectiveness of the proposed approach on both individual actions, interactions, multiview and large-scale datasets. In particular, we also indicate that the proposed method is invariant to viewpoint changes and requires low computational cost for training and inference. We hope that this study opens up a new door to exploit the big potential of skeletal data, which helps to address the current challenges in building real-world action recognition applications.

## Figures and Tables

**Figure 1 sensors-19-01932-f001:**
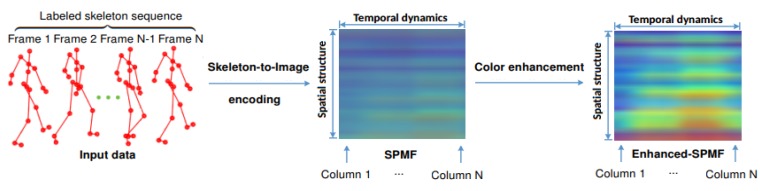
Overview of the proposed Enhanced-SPMF representation. Each skeleton sequence is transformed into a single RGB that is a motion map called SPMF [48]. A color enhancement technique [49] is then used to highlight the motion map and form the Enhanced-SPMF, which will be learned and classified by a deep learning model. Before computing the SPMF, a smoothing filter is adopted to reduce the effect of noise on skeletal data. Section 3 describes the details of the proposed approach.

**Figure 2 sensors-19-01932-f002:**
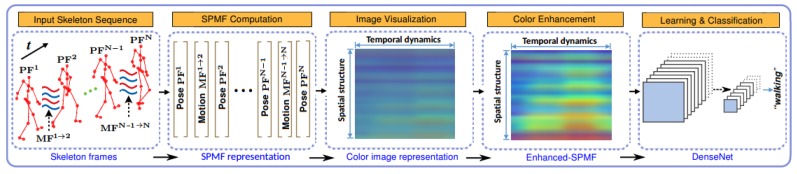
Schematic overview of the proposed approach. Each skeleton sequence is encoded in a single color image via a skeleton-based representation called SPMF. Each SPMF is built from pose vectors (PFs) and motion vectors (MFs) extracted from skeleton joints. They are then enhanced by an Adaptive Histogram Equalization (AHE) [49] algorithm and fed to a D-CNN for learning discriminative features and performing action classification. To achieve high-level learning performance during the training phase, we design and optimize different D-CNN models based on deep DenseNet [50], a recent state-of-the-art architecture for image recognition tasks.

**Figure 3 sensors-19-01932-f003:**
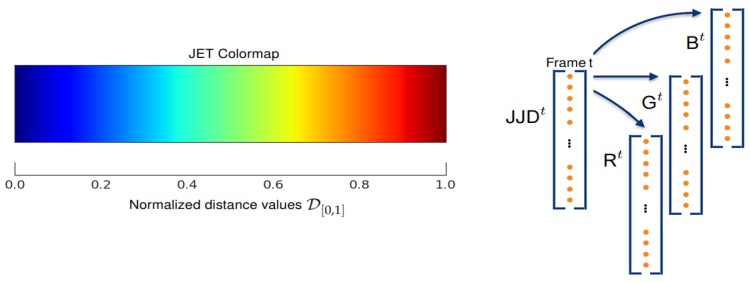
Illustration of the encoding process that converts joint-joint distance values to color points using a JET colormap.

**Figure 4 sensors-19-01932-f004:**
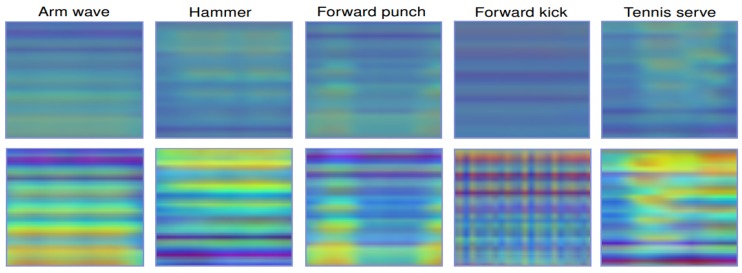
Results of the skeleton-to-image mapping process. The top row shows the proposed SPMF representations obtained from some samples of the MSR Action3D dataset [59]. The change in color reflects the change of distance and orientation between the joints. The bottom row shows generated images after applying the AHE algorithm [49].

**Figure 5 sensors-19-01932-f005:**
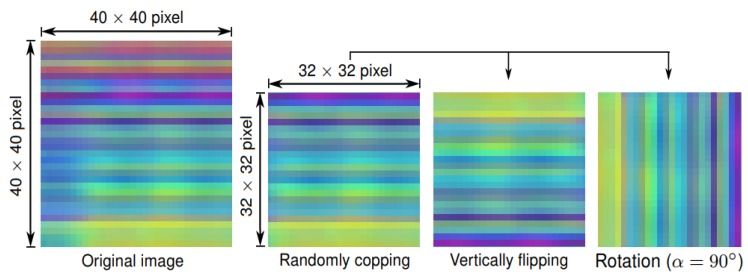
Illustration of data augmentation techniques used to generate more training samples.

**Figure 6 sensors-19-01932-f006:**
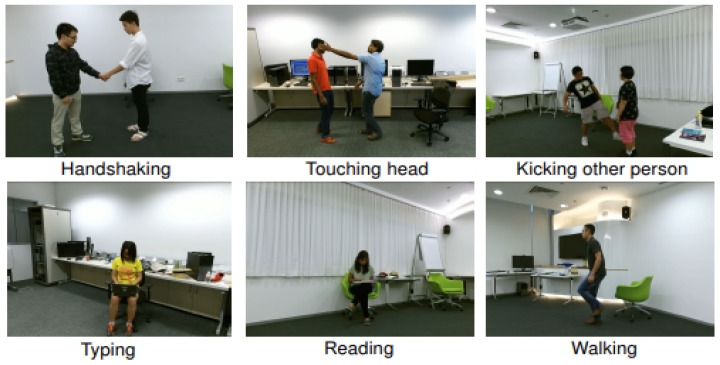
Some action classes of the NTU-RGB+D dataset [39]. Video samples have been captured by 3 Microsoft Kinect ${}^{\mathrm{TM}}$ v2 sensors concurrently at 30 FPS. The 3D skeletal data contains the three dimensional locations of 25 major body joints, at each frame. Figure was reproduced from the work of Shahroudy et al. [39] and used with permission from IEEE.

**Figure 7 sensors-19-01932-f007:**
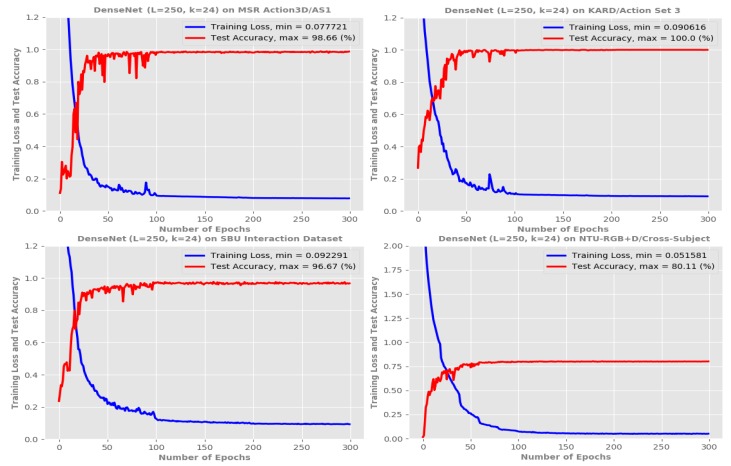
Training curves of the proposed DenseNet (*L* = 250, *k* = 24) on the MSR Action3D [59], KARD [60], SBU Kinect Interaction [61], and NTU-RGB+D [39] datasets. Almost all designed networks are able to reach the optimal weights after the first 100 epochs. The symbols *k* and *L* and denote the *“growth rate”* and the depth of the network, respectively.

**Figure 8 sensors-19-01932-f008:**
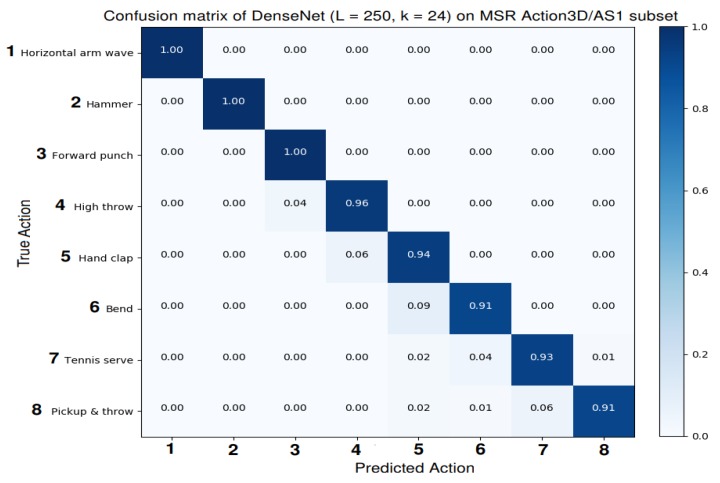
Confusion matrix of the proposed DenseNet (*L* = 250, *k* = 24) on the MSR Action3D/AS1 dataset. Ground truth action labels are on rows and predictions by the proposed method are on columns. We recommend the readers to use a computer and zoom in to see clearly these figures.

**Figure 9 sensors-19-01932-f009:**
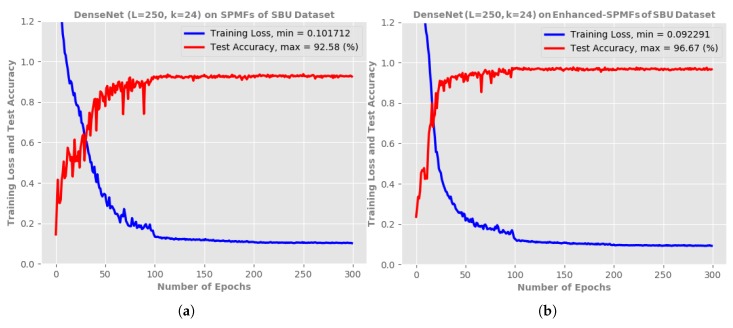
Training loss and test accuracy of the proposed DenseNet (L=100, k=12) on the SBU dataset [61]. (**a**) shows the obtained result when trained on SPMFs, while (**b**) reports the obtained result when trained on Enhanced-SPMFs. The symbols *k* and *L* denote the *“growth rate”* and the depth of the network, respectively.

**Figure 10 sensors-19-01932-f010:**
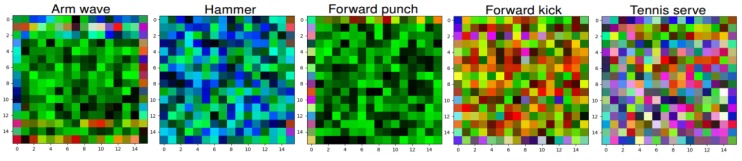
Visualization of feature maps learned by the proposed DenseNet (L=100, k=12) from several samples of the MSR Action3D dataset [59]. Best viewed in color.

**Figure 11 sensors-19-01932-f011:**
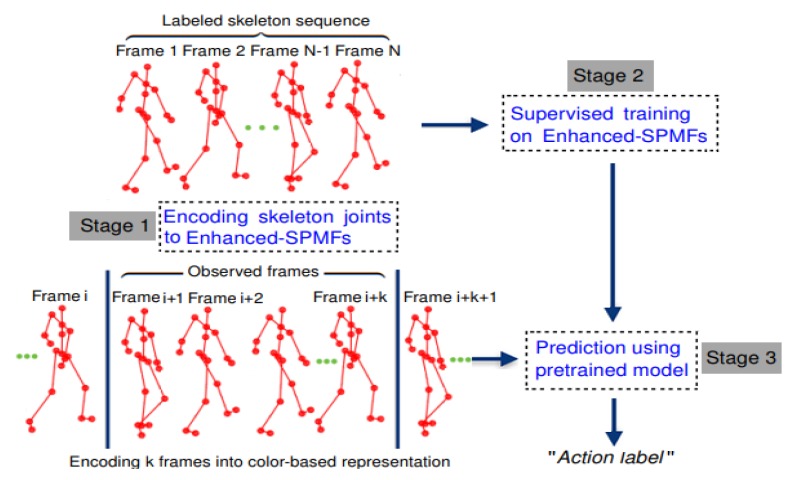
Three main stages of the proposed deep learning framework for recognizing human actions from skeleton sequences.

**Table 1 sensors-19-01932-t001:** Summary of advantages and disadvantages of previous methods and our proposed method.

Method & Authors	Data Modalities	Year	Advantages/Disadvantages
Bag of 3D Points [59]	Depth maps	2010	Simple and fast/Low accuracy, viewpoint dependent.
Lie Group Representation [23]	Skeletal data	2014	Robust to temporal misalignment and noise/Low accuracy.
Hierarchical LSTM [37]	Skeletal data	2015	Fast and high accuracy/ Easy to overfit.
Depth Motion Maps [69]	Depth maps	2016	Real-time latency/Low accuracy.
ST-LSTM Trust Gates [40]	Skeletal data	2016	View-invariant representation, robust to noise and occlusion/High computational cost.
Graph-Based Motion [36]	Skeletal data	2016	Robust to noise, high accuracy/Complex, parameter-dependent.
ST-NBNN [70]	Depth maps	2017	Simple and low computational cost/Parameter-dependent.
Ensemble TS-LSTM v2 [71]	Skeletal data	2017	High accuracy, robust to scale, rotation and translation/Data-hungry, high computational cost.
Bi-LSTM [72]	Skeletal data	2018	Invariant representation/Complex, low accuracy.
Our proposed method	Skeletal data	2019	View-invariant representation, real-time latency, high accuracy/Data-hungry, sensitive to data error of local fragments.

**Table 2 sensors-19-01932-t002:** The list of actions in three subsets AS1, AS2, and AS3 of the MSR Action 3D dataset [59].

AS1	AS2	AS3
*[a02] Horizontal arm wave*	*[a01] High arm wave*	*[a06] High throw*
*[a03] Hammer*	*[a04] Hand catch*	*[a14] Forward kick*
*[a05] Forward punch*	*[a07] Draw* x	*[a15] Side kick*
*[a06] High throw*	*[a08] Draw tick*	*[a16] Jogging*
*[a10] Hand clap*	*[a09] Draw circle*	*[a17] Tennis swing*
*[a13] Bend*	*[a11] Two hand wave*	*[a18] Tennis serve*
*[a18] Tennis serve*	*[a12] Forward kick*	*[a19] Golf swing*
*[a20] Pickup* & *Throw*	*[a14] Side-boxing*	*[a20] Pickup* & *Throw*

**Table 3 sensors-19-01932-t003:** The list of actions in three subsets of the KARD dataset [60].

Action Set 1	Action Set 2	Action Set 3
*Horizontal arm wave*	*High arm wave*	*Draw tick*
*Two-hand wave*	*Side kick*	*Drink*
*Bend*	*Catch cap*	*Sit down*
*Phone call*	*Draw tick*	*Phone call*
*Stand up*	*Hand clap*	*Take umbrella*
*Forward kick*	*Forward kick*	*Toss paper*
*Draw X*	*Bend*	*High throw*
*Walk*	*Sit down*	*Horizontal arm wave*

**Table 4 sensors-19-01932-t004:** Experimental results and comparison of the proposed method with state-the-art approaches on the MSR Action3D dataset [59]. The list is ordered by recognition performance, in which results that outperform previous works are in **bold**, while the best accuracies are in blue. Our previous work on SPMF [48] are marked in red.

Method (Protocol of [59])	Year	AS1	AS2	AS3	Aver.
Bag of 3D Points [59]	2010	72.90%	71.90%	71.90%	74.70%
Depth Motion Maps [69]	2016	96.20%	83.20%	92.00%	90.47%
Bi-LSTM [72]	2018	92.72%	84.93%	**97.89%**	91.84%
Lie Group Representation [23]	2014	95.29%	83.87%	98.22%	92.46%
FTP-SVM [72]	2018	95.87%	86.72%	**100.0%**	94.19%
Hierarchical LSTM [37]	2015	99.33%	94.64%	95.50%	94.49%
ST-LSTM Trust Gates [40]	2016	N/A	N/A	N/A	94.80%
Graph-Based Motion [36]	2016	93.60%	95.50%	95.10%	94.80%
ST-NBNN [70]	2017	91.50%	95.60%	97.30%	94.80%
ST-NBMIM [83]	2018	92.50%	95.60%	98.20%	95.30%
S-T Pyramid [84]	2015	99.10%	92.90%	96.40%	96.10%
Ensemble TS-LSTM v2 [71]	2017	95.24%	96.43%	**100.0%**	97.22%
SPMF Inception-ResNet-222 [48]	**2018**	**97.54%**	**98.73%**	**99.41%**	**98.56%**
Enhanced-SPMF DenseNet (*L* = 100, *k* = 12) (**ours**)	2018	**98.52**%	**98.66**%	99.09%	**98.76**%
Enhanced-SPMF DenseNet (*L* = 250, *k* = 24) (**ours**)	2018	**98.83**%	**99.06**%	99.40%	**99.10**%
Enhanced-SPMF DenseNet (*L* = 190, *k* = 40) (**ours**)	2018	**98.60**%	**98.87**%	99.36%	**98.94**%

**Table 5 sensors-19-01932-t005:** Average recognition accuracies (%) over three experiments A, B, and C and comparison with previous works on the KARD dataset [60]. The best accuracies are in blue. Results that surpass previous works are in **bold**.

Method (Protocol of [60])	Year	Acc. (%)
Hand-crafted Features [60]	2015	90.83%
Posture Feature+Multi-class SVM [85]	2016	97.20%
Key Postures+Multi-class SVM [86]	2016	99.30%
Enhanced-SPMF DenseNet (*L* = 100, *k* = 12) (**ours**)	2018	**99.74%**
Enhanced-SPMF DenseNet (*L* = 250, *k* = 24) (**ours**)	2018	**99.98%**
Enhanced-SPMF DenseNet (*L* = 190, *k* = 40) (**ours**)	2018	**99.88%**

**Table 6 sensors-19-01932-t006:** Action recognition accuracies (%) and comparison with previous works on the SBU Kinect Interaction dataset [61]. The best accuracies are in blue. Results that surpass previous works are in **bold**.

Method (Protocol of [61])	Year	Acc. (%)
Raw Skeleton [61]	2012	49.70%
Joint Features [61]	2012	80.30%
HBRNN [37] (reported in [91] )	2015	80.40%
CHARM [87]	2015	83.90%
Deep LSTM [41]	2017	86.03%
Joint Features [88]	2014	86.90%
ST-LSTM [40]	2016	88.60%
Co-occurrence+Deep LSTM [41]	2018	90.41%
STA-LSTM [89]	2017	91.51%
ST-LSTM+Trust Gates [40]	2018	93.30%
ST-NBMIM [83]	2018	93.30%
Clips+CNN+MTLN [90]	2017	93.57%
CNN Kernel Feature Map [96]	2018	94.36%
Two-stream RNN [91]	2017	94.80%
GCA-LSTM network [92]	2018	94.90%
Enhanced-SPMF DenseNet (*L* = 100, *k* = 12) (**ours**)	2018	**94.81%**
Enhanced-SPMF DenseNet (*L* = 250, *k* = 24) (**ours**)	2018	**96.67%**
Enhanced-SPMF DenseNet (*L* = 190, *k* = 40) (**ours**)	2018	**97.86%**

**Table 7 sensors-19-01932-t007:** Experimental results and comparison of the proposed method with previous approaches on the NTU-RGB+D dataset [39]. The best accuracies are in blue. Results that surpass previous works are in **bold**. Our previous works [48,51] are marked in red.

Method (Protocol of [39])	Year	Cross-Subject	Cross-View
Lie Group Representation [23]	2014	50.10%	52.80%
Hierarchical RNN [37]	2016	59.07%	63.97%
Dynamic Skeletons [93]	2015	60.20%	65.20%
Two-Layer P-LSTM [39]	2016	62.93%	70.27%
ST-LSTM Trust Gates [40]	2016	69.20%	77.70%
Skeleton-based ResNet [51]	**2018**	**73.40%**	**80.40%**
Geometric Features [74]	2017	70.26%	82.39%
Two-Stream RNN [91]	2017	71.30%	79.50%
Enhanced Skeleton [94]	2017	75.97%	82.56%
Lie Group Skeleton+CNN [95]	2017	75.20%	83.10%
CNN Kernel Feature Map [96]	2018	75.35%	N/A
GCA-LSTM [92]	2018	76.10%	84.00%
SPMF Inception-ResNet-222 [48]	**2018**	**78.89%**	**86.15%**
Enhanced-SPMF DenseNet (*L* = 100, *k* = 12) (**ours**)	2018	**79.31%**	**86.64%**
Enhanced-SPMF DenseNet (*L* = 250, *k* = 24) (**ours**)	2018	**80.11%**	**86.82%**
Enhanced-SPMF DenseNet (*L* = 190, *k* = 40) (**ours**)	2018	**79.28%**	**86.68%**

**Table 8 sensors-19-01932-t008:** Execution time of each stage of the proposed deep learning framework.

Stage	Average Processing Time (Second/Sequence)
1	$20.8\times 10^{-3}$ (Intel Core i7 3.2 GHz CPU)
2	0.164 (GTX 1080 Ti GPU)
3	$74.8\times 10^{-3}$ (CPU + GPU time)

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
