# Peer review of "Spatio–Temporal Image Representation of 3D Skeletal Movements for View-Invariant Action Recognition with Deep Convolutional Neural Networks"

_sensors, 2019, doi:10.3390/s19081932_

Reviewer 1 Report

this paper presents a novel skeleton-based representation and a deep learning framework for 3D
 action recognition using RGB-D sensors. Compared with there early publication in ICIP,  Enhanced-SPMF representation was proposed,and densenet was applied rather than Inception-ResNet .The method is designed reasonably and the results are clearly presented. .I recommend this paper to be published with minor revision. 

Before publishing, I concerned that the impact of resize the image representation to 32X32, since the body joints and frames are different. The second concern is why the Enhanced-SPMF is View-Invariant,not just experimentation

Author Response

Please see attached response.

Reviewer 2 Report

This paper proposed a method to recognize certain human actions. It generates an action map called skeleton posture-motion feature built from skeleton poses and their motions. Then, an adaptive histogram equalization algorithm is applied and eventually forms an enhanced action map, namely Enhanced-SPMF. The Deep Convolution Neural Networks based on the DenseNet architecture is exploited for learning and classification tasks. The proposed method is evaluated on four challenging datasets. The experimental results show good performance with time saving efficiency, and even outperformed most of the references in this paper. As a whole, this paper is presented adequately on both results and presentation. Nevertheless, there are still several comments and questions listed as follows.

1.      This paper is the extension of [1], written by the same authors. This paper enhanced SPMF presentation and achieved better performance. However, the improvement is very limited over [1], on average less than 1%. Since [1] was presented in ICIP 2018, an international conference, it makes sense to publish the extension version, i.e., this paper, in a journal. In addition, the short version, i.e., [1] did not explain the method in detail due to the limited space. Nevertheless, the authors should still clearly describe the improvement and additional contribution of this paper.

2.      In Fig. 2, PM++MF has overall 8 or 9 dimensions. How can we use a color map (RGB 3 dim) show the high dimensional image? Could the mapping from “SPMF representation” to “Color image representation” be explained more detailed?

3.      In Fig. 5, the rotation operation changes the time-space relationship to space-time, which seems making no sense for real data. Is it a good data augmentation technique in this method?

4.      Fig. 7, the font size is too small. There is plenty of space for larger fonts.

5.      Fig. 8, it used a lot of space to show similar figures. Maybe it is enough to show the most important results with less figures, similar to Fig. 9, if the space is a concern.

6.      The DenseNet used in this method is not described clearly, e.g., L, k, and the figures in Fig. 10.

Author Response

Please see attached response

Round  2

Reviewer 2 Report

The new edition has addressed all comments made by this reviewer and made enough modifications in the paper. I have no more questions. 

Author Response

Many thanks for your positive comment
